# Effect of Nitrogen Doping on the Photoluminescence of Amorphous Silicon Oxycarbide Films

**DOI:** 10.3390/mi10100649

**Published:** 2019-09-27

**Authors:** Jie Song, Rui Huang, Yi Zhang, Zewen Lin, Wenxing Zhang, Hongliang Li, Chao Song, Yanqing Guo, Zhenxu Lin

**Affiliations:** School of Materials Science and Engineering, Hanshan Normal University, Chaozhou 521041, China; songjie@hstc.edu.cn (J.S.); rhuang@hstc.edu.cn (R.H.); cavtor@126.com (Y.Z.); 2636@hstc.edu.cn (Z.L.); lhl4@hstc.edu.cn (H.L.); chaosong@hstc.edu.cn (C.S.); yqguo126@126.com (Y.G.)

**Keywords:** photoluminescence, amorphous silicon oxycarbide, nitrogen doping, defect, plasma enhanced chemical vapor deposition

## Abstract

The effect of nitrogen doping on the photoluminescence (PL) of amorphous SiC_x_O_y_ films was investigated. An increase in the content of nitrogen in the films from 1.07% to 25.6% resulted in red, orange-yellow, white, and blue switching PL. Luminescence decay measurements showed an ultrafast decay dynamic with a lifetime of ~1 ns for all the nitrogen-doped SiC_x_O_y_ films. Nitrogen doping could also widen the bandgap of SiC_x_O_y_ films. The microstructure and the elemental compositions of the films were studied by obtaining their Raman spectra and their X-ray photoelectron spectroscopy, respectively. The PL characteristics combined with an analysis of the chemical bonds configurations present in the films suggested that the switching PL was attributed to the change in defect luminescent centers resulting from the chemical bond reconstruction as a function of nitrogen doping. Nitrogen doping provides an alternative route for designing and fabricating tunable and efficient SiC_x_O_y_-based luminescent films for the development of Si-based optoelectronic devices.

## 1. Introduction

Efficient silicon (Si)-based luminescent materials are indispensable components to realize a cheap and complementary metal oxide semiconductor (CMOS) optical integration. Thus far, different systems of Si-based luminescent materials, such as SiO_x_, SiN_x_, SiC_x_, and SiN_x_O_y_, have been developed, and efforts have been devoted to understanding and ameliorating the light emission of Si-based materials [1,2,3,4,5,6,7,8,9]. Silicon oxycarbide (SiC_x_O_y_) has been widely explored because of its strong light emission and high solid solubility for rare earths [10,11,12,13]. SiC_x_O_y_ also features a tunable band gap. As such, it is beneficial to obtaining strong white electroluminescence at a low driving voltage in SiC_x_O_y_-based light-emitting diodes [14]. In the recent reference, Gallis et al. systematically studied the white photoluminescence (PL) dynamics from SiC_x_O_y_ film, where the band tail states related to the Si−O−C and/or the Si−C bonds were suggested as the sources of the luminescence [11]. Recently, optical gain was demonstrated in a-SiC_x_O_y_ under ultraviolet excitation, which was attributed to the formation of a three-level luminescence model with the intermediate level related to Si dangling bond (DB) defects radiative state [15]. Furthermore, an increase in C content in SiC_x_O_y_ films can cause a strong light emission ranging from near-infrared to orange regions [15]. Although performance is enhanced in SiC_x_O_y_ films, progress remains slow. The main obstacle lies in the fact that the light emission efficiency generally remains too low to allow the fabrication of efficient light-emitter devices. To date, studies on the effect of doping on the optical properties of SiC_x_O_y_ films have mainly focused on rare earth (RE) doping, such as Er and Eu doping [16,17,18]. However, up to now, the effect of other elements on the optical properties of SiC_x_O_y_ films is still unclear.

In this letter, the effect of nitrogen doping on the PL of amorphous SiC_x_O_y_ film was investigated. Interestingly, an increase in nitrogen content in the films induced strong red, orange-yellow, white, and blue switching PL. Combining the PL results with the analysis of the microstructure and the chemical bonding configurations within the films, it suggests that the rearrangement of chemical bonds with varying nitrogen plays an important role in the evolution of PL characteristics in the films.

## 2. Materials and Methods

Nitrogen-doped SiC_x_O_y_ films with the thickness of 550 nm were grown at 250 °C on Si substrates and quartz by radio frequency (RF) glow-discharge decomposition of SiH_4_, CH_4_, O_2_, and NH_3_ mixtures in the very high frequency plasma enhanced chemical vapor deposition (VHF-PECVD) system. The flow rates of SiH_4_, CH_4_, and O_2_ were kept at 3.5, 5, and 1.2 sccm, respectively, whereas the flow rate of NH_3_ varied from 0.5 sccm to 5 sccm to control the N content in the films. The films were named S_x_ (x = 1, 2, 3, 4) for the NH_3_ flow rates at 0.5, 1, 3, and 5 sccm, respectively. The RF power and the deposition pressure for the growth were maintained at 30 W and 20 Pa, respectively. The optical band gaps of the films were calculated using the Tauc method, which were determined by spectrophotometer (Shimadzu UV-3600, Shimadzu Corporation, Kyoto, Japan). The PL spectra of the films were measured at room temperature by use of a fluorescence spectrophotometer (Jobin Yvon fluorolog-3, Horiba, Ltd., Kyoto, Japan). Time resolved PL were measured by use of an Edinburgh FLS1000 spectrometer (Edinburgh Instruments Ltd., Livingston, UK) equipped with a 600 mW 375 nm-laser beam with the repetition rate of 20 MHz. The microstructures of the films were characterized by Raman spectra. The chemical bonds of the films were examined by Fourier transform infrared (FTIR) absorption, and Si, O, C, and N contents in the films were identified through X-ray photoelectron spectroscopy.

## 3. Results and Discussion

Figure 1a shows the PL spectra of the SiCO:N films prepared at different NH_3_ flow rates. Strong visible light emission could be tuned from red to blue regions at room temperature by adjusting the NH_3_ flow rates increased from 0.5 sccm to 5 sccm. Red, orange-yellow, white, and blue switching luminescence were strong enough to be seen with naked eyes even at 325 nm Xe lamp light excitation. Figure 1b illustrates the optical band gap energy of the films as a function of the NH_3_ flow rate. The optical band gap *E_opt_* of the films was obtained in accordance with the Tauc plot Equation (1):(1)$$\left(\alpha h\nu {)}^{1/2}=A^{1/2}(h\nu -E_{opt}\right)$$
where *α* is the absorption coefficient, *A* is a coefficient quantifying the slope of the absorption edge, and *hν* is the photon energy [19]. The calculated *E_opt_* increases linearly from 2.83 eV to 3.66 eV as the NH_3_ flow rate increases from 0.5 sccm to 5 sccm. This finding demonstrates that N doping can widen the bandgap of SiC_x_O_y_ films, which may result from the substitution of stronger Si–N bonds for weak Si–Si bonds or Si–C bonds. The comparison of PL with *E_opt_* results indicated that the value of PL peak energy of all the films was obviously smaller than the corresponding *E_opt_*, suggesting that the origin of PL was not from the band-to-band recombination.

The microstructure of the SiCO:N films was examined using Raman spectra (Figure 2a) to further understand the origin of PL characteristics. All the SiCO:N films exhibited similar line shape characteristics typical of amorphous silicon-based materials. A broad Raman band, which was ascribed to the transverse optical (TO) vibration mode of amorphous silicon, peaked at ~470 cm^−1^ for all the SiCO:N films. These results showed that all the SiCO:N films had a uniform amorphous structure without the presence of Si nanocrystals [20]. Furthermore, there was no obvious change in the surface morphology of the films prepared at different NH_3_ flow rates, as was revealed by atomic force microscopy (Figure 2b).

The films were measured by time-resolved PL to obtain further insights into the PL mechanism of the SiCO:N films (Figure 3). The decay curves could be fitted with a double exponential function:(2)$$I(t)=A_{1}\exp(\frac{-t}{\tau_{1}})+A_{2}\exp(\frac{-t}{\tau_{2}})$$
where *A*_i_ and *τ*_i_ (i = 1, 2) are the normalized amplitudes of the components and the time constants, respectively [21]. The obtained average lifetime of the SiCO:N films was about 1 ns. The luminescent dynamic behavior was similar to that observed in defect-related luminescent Si-based materials, such as SiN_x_O_y_ and SiC_x_O_y_ [19,21]. Furthermore, it was also found that the luminescence decay lifetimes in our case were shorter than those in the band-tail recombination model where a broader band-tail brought a longer lifetime, as the photogenerated carriers could be thermalized into deeper localized states [22]. Therefore, the results suggested that the light emission of the SiCO:N films originated from the defect luminescent centers in the films.

The FTIR spectra of the SiCO:N films were obtained to study the local bonding changes in the films grown at different NH_3_ flow rates (Figure 4). In the S1 film, the vibration modes related to Si–C, Si–N, C–Si–O, Si–H, and C–H bonds could be clearly observed. The bands centered at 860 and 1039 cm^−1^ could be ascribed to Si–N and C–Si–O stretching modes, respectively [23,24]. Additionally, a band at 1265 cm^−1^ was assigned to the Si–CH_3_ stretching vibration [25]. A small band shoulder at 800 cm^−1^ was observed and was assigned to the Si–C stretching vibration [24]. A distinct absorption peak at 2170 cm^−1^ and a weak band located at 2965 cm^−1^ were attributed to the Si–H and the C–H stretching vibrations, respectively [26]. A weak band around 3375 cm^−1^ was associated with the N–H stretching mode [23]. The most important feature for the FTIR spectra was the strong dependence of major bands on NH_3_ flow rates. As the NH_3_ flow rates increased, the intensity of C–Si–O bonds gradually decreased, and the peak gradually became red shifted. As the NH_3_ flow rate increased to 3 sccm, this band broadened and red shifted to ~1010 cm^−1^ with a shoulder at ~940 cm^−1^, which was assigned to the N–Si–O vibration [27]. As the NH_3_ flow rate further increased to 5 sccm, the band of the N–Si–O vibration became dominant, indicating that the silicon oxycarbide-dominant phase of the film transformed into silicon oxynitride. Apparently, the increase in the NH_3_ flow rate resulted in chemical bond reconstruction in the films. Based on the FTIR spectra, the evolution of PL characteristics could be suggested from the chemical bond reconstruction in SiCO:N films.

The composition of the SiCO:N films was examined through X-ray photoelectron spectroscopy (XPS) (Figure 5). The atomic percentages of Si, C, O, and N in the SiCO:N film fabricated at an NH_3_ flow rate of 0.5 sccm were 51.89%, 19.82%, 27.22%, and 1.07%, respectively. This finding indicated that the Si-rich silicon oxycarbide phase was dominant in the S1 film. The change in the XPS spectra was the gradual decrease in Si and C concentrations with the increase in N concentration and NH_3_ flow rates (Figure 5). As the NH_3_ flow rate increased to 5 sccm, the N concentration rapidly increased to 25.6%, whereas the Si and the C concentrations decreased to 40.8% and 8.22%, respectively. This finding was consistent with the observed results in the FTIR spectra shown in Figure 4, that is, the N–Si–O vibration band became dominant, while the C–Si–O vibration band significantly weakened as the NH_3_ flow rate increased to 5 sccm. This result indicated that the dominant phase in the films changed from silicon oxycarbide to silicon oxynitride when the NH_3_ flow rate increased to 5 sccm.

The PL decay analysis (Figure 3) revealed that the luminescent dynamic behavior in the nitrogen doped SiC_x_O_y_ films featured a defect-related luminescent characteristic, as observed in SiN_x_O_y_ and SiC_x_O_y_ films. Previous studies clarified that C-related nonbridging oxygen hole centers (NBOHC) are the principal radiative recombination centers in silicon oxycarbide, and they are responsible for light emission ranging from the green region to the red region [28]. In our case, the PL intensity in the film decreased as the NH_3_ flow rate increased. This change was similar to that of the intensity of the C–Si–O bonds (Figure 4). Therefore, the observed tunable light emissions from green to red may have originated from recombination through C-related NBOHC defects in SiC_x_O_y_ films. The PL spectra of the S3 film could be deconvoluted into a strong green band and a weak blue band. As the NH_3_ flow rate increased to 5 sccm, the intensity of the green PL band decreased dramatically, whereas the blue PL band of the film S4 became dominant. This behavior could be attributed to the change in the dominant phase of the films from silicon oxycarbide to silicon oxynitride as a result of the increase in NH_3_ flow rate to 5 sccm (Figure 4). In the case of amorphous SiN_x_O_y_ films, the blue PL could be ascribed to the radiative recombination between N–Si–O defect states and the valence band tail states [27]. Thus, the blue PL from S3 and S4 was likely from N–Si–O defect luminescent centers.

## 4. Conclusions

In summary, we report the effect of nitrogen doping on the PL of amorphous SiC_x_O_y_ films. Nitrogen doping can induce strong red, orange-yellow, white, and blue switching PL with a recombination lifetime in nanoseconds. This process can also widen the band gap of SiC_x_O_y_ films. The PL results and the FTIR analyses reveal that the switching characteristics in PL originate from the variation in defect luminescent centers resulting from the chemical bond re-construction as a function of nitrogen doping. Apparently, nitrogen doping provides an alternative route for designing and fabricating tunable and efficient SiC_x_O_y_-based luminescent films for the development of Si-based optoelec-tronic devices.

## Figures and Tables

**Figure 1 micromachines-10-00649-f001:**
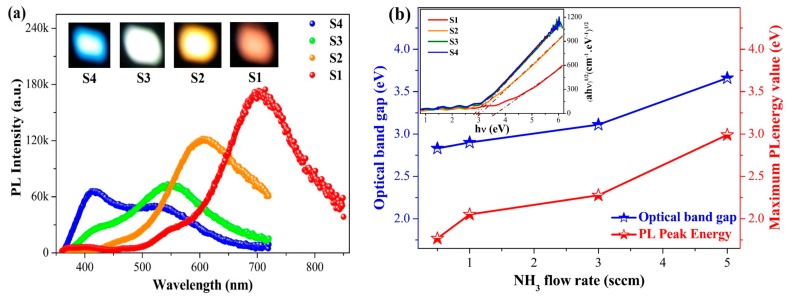
(**a**) Photoluminescence (PL) spectra of the SiCO:N films prepared by different NH_3_ flow rates: S1 (0.5 sccm), S2 (1 sccm), S3 (3 sccm), and S4 (5 sccm). The inset is the optical images of PL from the films under 325 nm Xe lamp light excitation. (**b**) The optical band gap of the SiCO:N films vs. the NH_3_ flow rates. The inset shows the ((α*hν*)^1/2^·vs·*hν*) plot of the SiCO:N film S_x_ (x = 1, 2, 3, 4).

**Figure 2 micromachines-10-00649-f002:**
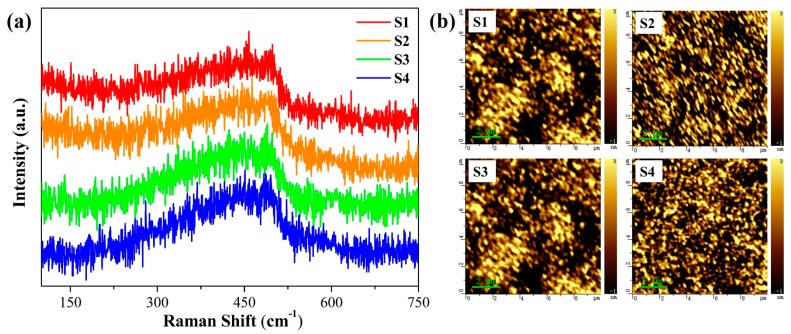
(**a**) Raman spectra of the SiCO:N films with various NH_3_ flow rates, (**b**) atomic force microscopic images of the SiCO:N films prepared by different NH_3_ flow rates: S1 (0.5 sccm), S2 (1 sccm), S3 (3 sccm), and S4 (5 sccm).

**Figure 3 micromachines-10-00649-f003:**
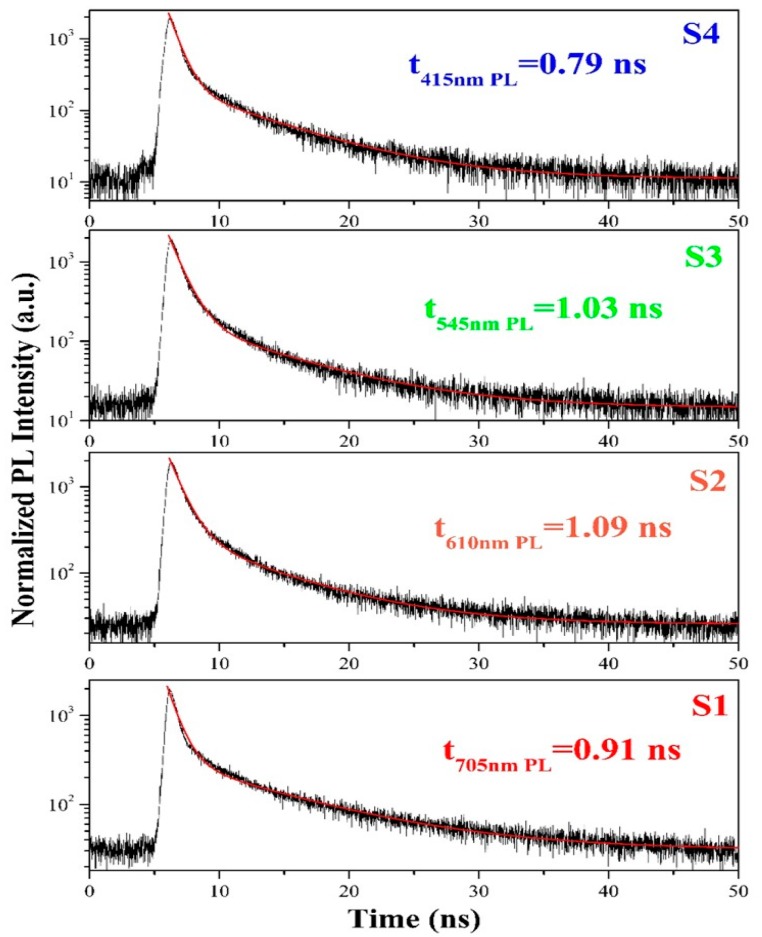
Room temperature time resolved photoluminescence for the SiCO:N films with various NH_3_ flow rates.

**Figure 4 micromachines-10-00649-f004:**
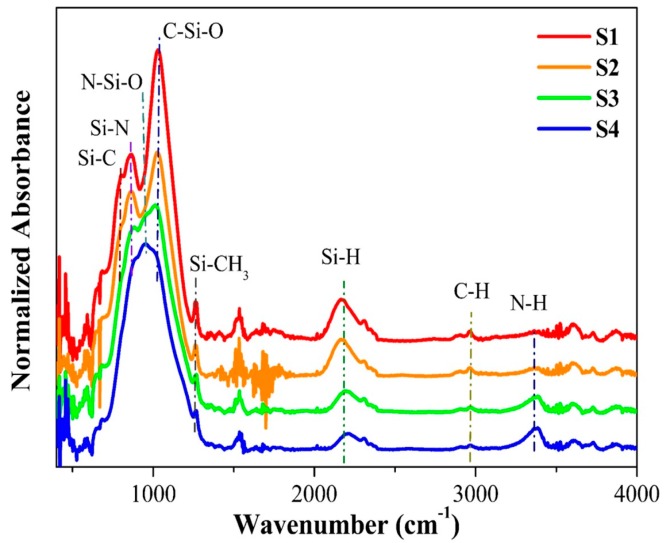
Fourier transform infrared (FTIR) spectra of the SiCO:N films grown at different NH_3_ flow rates.

**Figure 5 micromachines-10-00649-f005:**
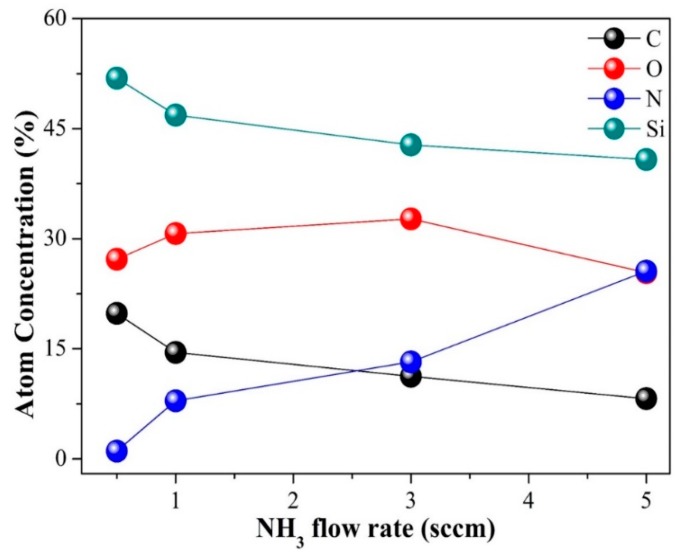
The atom concentration of Si, C, O, and N of the SiCO:N films against the NH_3_ flow rates.

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
