# Peer review of "Effect of Nitrogen Doping on the Photoluminescence of Amorphous Silicon Oxycarbide Films"

_micromachines, 2019, doi:10.3390/mi10100649_

Round 1
Reviewer 1 Report
Please consider the following comments:
1) Enhance the Introduction so the motivation for your work is better elaborated and stated (other than the sentence in lines 38-39). For example, try to answer the question, what's the driving force behind investigating N doping in these films?
2) Add more details on the experimental parameters for the PL and especially the time-resolved PL measurements (e.g. experimental conditions, excitation power, laser source, repetition rate, PL system response time) and the way the PL decays were analyzed.
3) In the caption of Figure 1B, add the description for the PL data shown. Furthermore, add the rest of the absorption spectra in the inset (only S1 is shown). This way the statement in the manuscript that an increase in N concentration in SiCO films leads to an increase in their optical bandgap can be more justifiable.
To this end, add reference in line 80.
4) Fix wording in lines 93-94; "were subjected to" is not the proper way to state this, probably "were measured by time-resolved PL".
5) Elaborate more and justify the statement in line 100-101, "results suggest that the light emission of the SiCO:N films originates from the defect luminescent centers in the films". It is written in an extremely simplistic way.
6) Lines 124-125: you may want to correlate the PL data (integrated PL or PL peak intensity) to the FTIR findings (C-Si-O absorption coefficient; thickness corrected absorbance). This can be seen in conjunction with comment 5.
7) Line 134: Please consider removing "remarkable feature".
8) Lines 155-156: Ref. 26 is not adequate for this statement. Needs different reference and rewording.
Hope this helps!
Author Response
Response to the comments of Editor and Reviewer
We sincerely appreciate the Editor’s comments and suggestions about our manuscript. According to your requirements, we have carefully corrected the technical issues in the revised manuscript. Following is our point-by-point reply.
Reviewer: 1
1) Enhance the Introduction so the motivation for your work is better elaborated and stated (other than the sentence in lines 38-39). For example, try to answer the question, what's the driving force behind investigating N doping in these films?
Response:
According to the Reviewer suggestion, we have added the research motivation in the introduction section (Please see Page1, line38-40).
2) Add more details on the experimental parameters for the PL and especially the time-resolved PL measurements (e.g. experimental conditions, excitation power, laser source, repetition rate, PL system response time) and the way the PL decays were analyzed.
Response:
According to the Reviewer suggestion, we have added the information about the experimental parameters for the PL and the time-resolved PL measurements and the way the PL decays were analyzed (Please see Page2, line60-62 and Page3, line103-104).
3) In the caption of Figure 1B, add the description for the PL data shown. Furthermore, add the rest of the absorption spectra in the inset (only S1 is shown). This way the statement in the manuscript that an increase in N concentration in SiCO films leads to an increase in their optical bandgap can be more justifiable.
To this end, add reference in line 80.
Response:
According to the Reviewer suggestion, we have added the information about the description for the PL data and the rest of the absorption spectra in the inset of Figure 1 (b) in the revised manuscript (Please see Page2, line67). In addition, we have also cited the corresponding literatures [19] in the revised manuscript (Please see Page3, line83).
4) Fix wording in lines 93-94; "were subjected to" is not the proper way to state this, probably "were measured by time-resolved PL".
Response:
We have revised the sentence as "The films were measured by time-resolved PL to obtain further insights into the PL mechanism of the SiCO:N films” (Please see Page3, line103-104)
5) Elaborate more and justify the statement in line 100-101, "results suggest that the light emission of the SiCO:N films originates from the defect luminescent centers in the films". It is written in an extremely simplistic way.
Response:
According to the suggestion, we have added more detailed information to suggest that the light emission of the SiCO:N films originates from the defect luminescent centers in the films (Please see Page4, line109-111).
6) Lines 124-125: you may want to correlate the PL data (integrated PL or PL peak intensity) to the FTIR findings (C-Si-O absorption coefficient; thickness corrected absorbance). This can be seen in conjunction with comment 5.
Response:
According to the suggestion, a comparison between the PL intensity and the intensity of the C-Si-O bond has been made.
Please see Page 6 line 159-160: “In our case, the PL intensity in the film decreases as the NH3 flow rate increases. This change is similar to that of the intensity of the C–Si–O bonds (Figure 4)”.
7) Line 134: Please consider removing "remarkable feature".
Response:
We agree with the reviewer and have revised “the remarkable feature” as “the change” (Please see Page5, line146).
8) Lines 155-156: Ref. 26 is not adequate for this statement. Needs different reference and rewording.
Response:
According to the suggestion, we have added the corresponding literatures [27] in the revised manuscript (Please see Page7, line251).

Reviewer 2 Report
The author studied the effect of Nitrogen plasma during SiCO film growth on the PL characteristics. Experimental design and results are reasonable and explanation is also scientific. So, this manuscript can be published as current form after adding minor isssue as written in below;
The author did not mentioned about the thickness of SiCO films. How much the thickness of each films. Also add about the cross-sectional SEM image to prove the thickness and add comment about the film roughness (using AFM or SEM images)
Author Response
Response to the comments of Editor and Reviewer
We sincerely appreciate the Editor’s comments and suggestions about our manuscript. According to your requirements, we have carefully corrected the technical issues in the revised manuscript. Following is our point-by-point reply.
Reviewer: 2
The author studied the effect of Nitrogen plasma during SiCO film growth on the PL characteristics. Experimental design and results are reasonable and explanation is also scientific. So, this manuscript can be published as current form after adding minor isssue as written in below;
The author did not mentioned about the thickness of SiCO films. How much the thickness of each films. Also add about the cross-sectional SEM image to prove the thickness and add comment about the film roughness (using AFM or SEM images)
Response:
Figure 1 The transmission spectra of the SiCO:N films prepared by different NH3 flow rates: S1(0.5sccm), S2(1sccm), S3(3sccm), and S4(5sccm).
Figure 2 Atomic force microscopic images of the SiCO:N films prepared by different NH3 flow rates: (a) S1(0.5sccm), (b) S2(1sccm), (c) S3(3sccm), and (d) S4(5sccm).
According to the suggestion, we have added the information about the thickness of SiCO:N films in the Materials and Methods (Please see Page2, line51). In our case, the thickness of SiCO:N films were are determined by the Swanepoel envelope method which is based on the optical transmission spectra measured in the spectral range 200-1600 nm, as shown in Figure 1.
According to the suggestion, the SiCO:N films prepared at different NH3 flow rates are characterized by using AFM. Figure 2 displays the image of the SiCO:N films prepared by different NH3 flow rates: (a) S1(0.5sccm), (b) S2(1sccm), (c) S3(3sccm), and (d) S4(5sccm). As shown in Figure 2, with increasing NH3 flow rate up to 5sccm, the surface morphology of the film almost do not change. We have added the corresponding description into the revised manuscript: Furthermore, there is no obvious change in the surface morphology of the films prepared at different NH3 flow rates, as is revealed by atomic force microscopy (Please see Page3, line94-96).

Reviewer 3 Report
I have carefully read and revise the manuscript, "Effect of Nitrogen Doping on the Photoluminescence of Amorphous Silicon oxycarbide Films", authored by Song et al.
The work is of an average category but can be improved significantly. At the present form, it is not suitable for publication.
I have the following concerns related to this work, which must be addressed in its revised form and should be incorporated in the manuscript:
The introduction does not present the advances in this field in detail. What are the advances in understanding those physical behaviors? Why does the lifetime change with Nitrogen doping? Give inherent reasons. The surface of the films has not been analyzed. Fig 4: Si-N, N-Si-O bands decrease upon Nitrogen doping, why? One scheme must be provided to directly correlate with the change in luminescence intensity. The study must illustrate physical insights rigorously. Fig. 1b: right-side scale: change the title as "maximum energy of the emitted light" or a similar one. Use the same set of colors in all the figures to identify the S1-S4 samples.So, the manuscript must be revised for considering a further decision.
Author Response
Response to the comments of Editor and Reviewer
We sincerely appreciate the Editor’s comments and suggestions about our manuscript. According to your requirements, we have carefully corrected the technical issues in the revised manuscript. Following is our point-by-point reply.
Reviewer: 3
I have carefully read and revise the manuscript, "Effect of Nitrogen Doping on the Photoluminescence of Amorphous Silicon oxycarbide Films", authored by Song et al.
The work is of an average category but can be improved significantly. At the present form, it is not suitable for publication.
I have the following concerns related to this work, which must be addressed in its revised form and should be incorporated in the manuscript:
1) The introduction does not present the advances in this field in detail. What are the advances in understanding those physical behaviors?
Response:
According to the Reviewer suggestion, we have added the research motivation in the introduction section (Please see Page1, line38-40).
2) Why does the lifetime change with Nitrogen doping? Give inherent reasons.
Response:
We sincerely thank the reviewer for his/her valuable suggestion. However, in our case the obtained average lifetime of all the SiCO:N films is about 1 ns. The PL lifetime of the SiCO:N films have little change as the increase of the NH3 flow rate. We still cannot clarify the origin of the PL lifetime change based on the above experimental data. We would continually investigate it in the future.
3) The surface of the films has not been analyzed.
Response:
Figure 1 Atomic force microscopic images of the SiCO:N films prepared by different NH3 flow rates: (a) S1(0.5sccm), (b) S2(1sccm), (c) S3(3sccm), and (d) S4(5sccm).
According to the suggestion, the SiCO:N films prepared at different NH3 flow rates are characterized by using AFM. Figure 2 displays the image of the SiCO:N films prepared by different NH3 flow rates: (a) S1(0.5sccm), (b) S2(1sccm), (c) S3(3sccm), and (d) S4(5sccm). As shown in Figure 1, with increasing NH3 flow rate up to 5sccm, the surface morphology of the film almost do not change. We have added the corresponding description into the revised manuscript: Furthermore, there is no obvious change in the surface morphology of the films prepared at different NH3 flow rates, as is revealed by atomic force microscopy (Please see Page3, line94-96).
4) Fig 4: Si-N, N-Si-O bands decrease upon Nitrogen doping, why? One scheme must be provided to directly correlate with the change in luminescence intensity. The study must illustrate physical insights rigorously.
Response:
We sincerely thank the reviewer for his/her valuable suggestion. However, in our case the intensity of the N–Si–O vibration band gradually increase while the C–Si–O vibration band significantly weakens as the increase of the NH3 flow rate. These results indicate that the C–Si–O bonds transformed into N–Si–O bonds as result of the increase in N concentration and NH3 flow rates. Comparing the FTIR spectra with PL spectra, it is interesting to find that the behavior of green and blue emission intensity with the NH3 flow rate shown in Figure 1(a) and Figure 4 exhibited a similar trend to that of the intensity of C–Si–O and N–Si–O bonds, respectively. This suggested that the evolution of PL characteristics is strongly correlated with the chemical bonds reconstruction in the film. The above results are shown and discussed in the revised manuscript (Please see Page 6, line 159-168).
5) Fig. 1b: right-side scale: change the title as "maximum energy of the emitted light" or a similar one.
Response:
According to the suggestion, we have revised the title in Fig. 1(b) (Please see Page 2, line 67).
6) Use the same set of colors in all the figures to identify the S1-S4 samples.
So, the manuscript must be revised for considering a further decision.
Response:
According to the suggestion, we have revised all the figure in the revised manuscript.

Round 2
Reviewer 3 Report
In the revised version all the concerns have been addressed well except the comment 1, i.e.: "The introduction does not present the advances in this field in detail. What are the advances in understanding those physical behaviors?"
The work can be accepted after revising the manuscript, according to comment 1.
Author Response
Response to the comments of Editor and Reviewer
We sincerely appreciate the Editor’s comments and suggestions about our manuscript. According to your requirements, we have carefully corrected the technical issues in the revised manuscript. Following is our point-by-point reply.
Reviewer: 3
I have carefully read and revise the manuscript, "Effect of Nitrogen Doping on the Photoluminescence of Amorphous Silicon oxycarbide Films", authored by Song et al.
The work is of an average category but can be improved significantly. At the present form, it is not suitable for publication.
I have the following concerns related to this work, which must be addressed in its revised form and should be incorporated in the manuscript:
1) The introduction does not present the advances in this field in detail. What are the advances in understanding those physical behaviors?
Response:
According to the Reviewer suggestion, we have added the information about the advances in understanding the PL mechanism of SiCxOy film in the introduction section (Please see Page1, line34-36).
